# Functional space analyses reveal the function and evolution of the most bizarre theropod manual unguals

Zichuan Qin [1✉], Chun-Chi Liao[2], Michael J. Benton [1✉] & Emily J. Rayfield [1✉]

Maniraptoran dinosaurs include the ancestors of birds, and most used their hands for grasping and in flight, but early-branching maniraptorans had extraordinary claws of mysterious function. Alvarezsauroids had short, strong arms and hands with a stout, rock-pick-like, single functional finger. Therizinosaurians had elongate fingers with slender and sickle-like unguals, sometimes over one metre long. Here we develop a comprehensive methodological framework to investigate what the functions of these most bizarre bony claws are and how they formed. Our analysis includes finite element analysis and a newly established functional-space analysis and also involves shape and size effects in an assessment of function and evolution. We find a distinct functional divergence among manual unguals of early-branching maniraptorans, and we identify a complex relationship between their structural strength, morphological specialisations, and size changes. Our analysis reveals that efficient digging capabilities only emerged in late-branching alvarezsauroid forelimbs, rejecting the hypothesis of functional vestigial structures like *T. rex*. Our results also support the statement that most therizinosaurians were herbivores. However, the bizarre, huge *Therizinosaurus* had sickle-like unguals of such length that no mechanical function has been identified; we suggest they were decorative and lengthened by peramorphic growth linked to increased body size.

[1] School of Earth Sciences, Life Sciences Building, University of Bristol, Tyndall Avenue, Bristol BS8 1TQ, UK. [2] Key Laboratory for the Evolutionary Systematics of Vertebrates, Institute of Vertebrate Paleontology and Paleoanthropology, Chinese Academy of Sciences, Beijing 100044, China. ✉email: zichuan.qin@bristol.ac.uk; mike.benton@bristol.ac.uk; e.rayfield@bristol.ac.uk

Our understanding of maniraptoran dinosaurs has dramatically improved in the past three decades benefiting from remarkable, well-preserved fossils[1]. Late-branching maniraptorans, especially the ancestors and kin of birds, have been well investigated not only in morphology and phylogeny[1,2], but also in functional evolution and ecology[1,3]. Early-branching maniraptorans, like alvarezsauroids and therizinosaurians, show bizarre morphological characters and they occupied enigmatic ecological niches[4,5], but their likely functions have not been studied in detail. Therizinosaurians and alvarezsauroids, represent early maniraptoran branches in recent phylogenies[6–9]. Some therizinosaurians evolved large body sizes, with highly elongate forelimbs and sickle-like unguals[10–12], whilst alvarezsauroids underwent rapid miniaturisation to become the smallest non-avian dinosaurs ever[13], but with shortened forelimbs and enlarged rock-pick-like unguals[14,15] (Fig. 1a, b).

For both clades, there are numerous functional and behavioural hypotheses mostly based on morphological evidence and functional analyses of the skulls, lower jaws[4,5,16,17] and hips[18]. Previous studies have already discussed their bizarre forelimbs, especially their exaggerated sizes when compared to overall body dimensions[14,19], strange anatomies[14,19], morphological evolution[12,20], and some functional simulations using finite-element analysis[12]. Therizinosaurians are better understood, and they are assumed to have been giant bipedal, ground sloth-like herbivorous animals[5,12,21]. With reference to the most remarkable elongate sickle-like unguals that emerged in late-branching members, simulation-based research suggested they were most optimal for hook-and-pull functions (defined here as looping the claw tip around and behind an object, then pulling), and possibly also for cutting tree branches[12]. On the other hand, palaeobiological hypotheses of the single functional finger ungual from late-branching alvarezsauroids are mainly based on

morphological and anatomical studies[14,15,22], and there has been a little quantitative investigation. Prevailing assumptions based on anatomy suggest that late-branching alvarezsauroids were insectivorous and forelimb hook-and-pull or scratch diggers[14,15,22], supported by their miniaturisation and highly specialised manus, but an alternative view is that they were egg-eaters that used their specialised unguals to remove egg shells[23].

The most distinctive characteristics of the forelimbs of therizinosaurians and alvarezsauroids are their highly modified, proportionally enlarged distal autopodia, namely their narrow or robust bony unguals. As the tips of the mechanical system of forelimbs, and the bony parts which bear the most stress, these manual unguals are key to revealing forelimb functions. However, compared to the cases of other highly specialised jaws, axial skeletons, or antlers in extinct species[24–26], there has been little research on the specific forming process of the manual claws of these two most distinctive maniraptorans. One issue is that studies on the morphology and function of unguals of maniraptoran dinosaurs have so far been based on two approaches, with morphological analyses conducted mainly on two-dimensional lateral shapes[27,28] and functional simulations focusing mostly on three-dimensional models[29,30]. These two approaches have made it difficult to discuss morphology and function within a common framework. In some cases at least, morphological specialisation did not match functional performance[12]. Therefore, a comprehensive method, which can quantitatively analyse and compare these two types of data, is required to solve this problem.

We present a new method called functional-space analysis (FSA), based on the previously published 'intervals' method, a method to quantify FE simulation results[31]. In our FSA pipeline (Fig. 1c–f), we establish 'functional hulls' by calculating and visualising the FE simulation results classed according to different simulations or specimens, and the classic Euclidean geometry

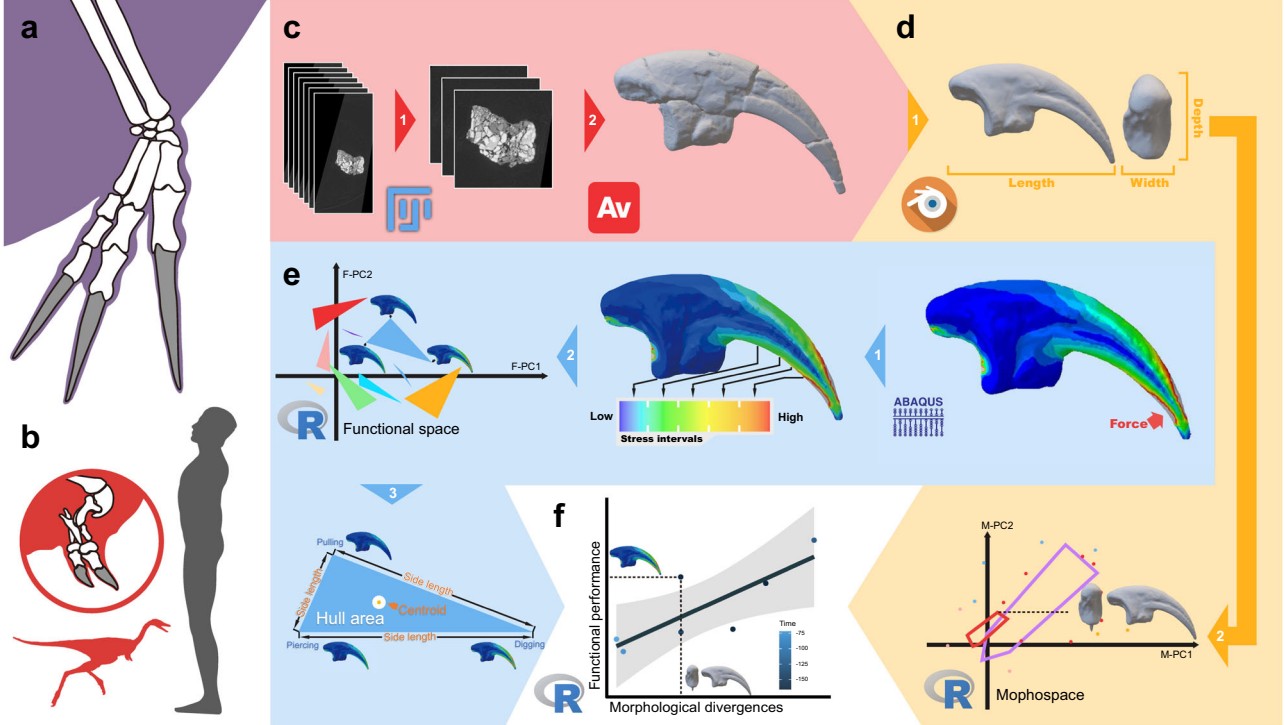

**Fig. 1 Key taxa and work pipeline use in this paper.** Silhouettes show the large and elongated forelimb of the late-branching therizinosaurian *Therizinosaurus* (**a**) and the overall body shape and highlighted forelimb of the late-branching alvarezsauroid *Mononykus* (**b**), scaled against an adult human (height ~1.8 m). The work pipeline demonstrated by an ungual model from the Jurassic alvarezsauroid *Haplocheirus*, includes processes of 3D model reconstruction (**c**); model smoothing, measurement, and morphological analysis (**d**); finite-element analysis, 'intervals' method and functional-space analysis (**e**); and total evidence functional assessment (**f**).

parameters of these functional spaces display functional divergence within objects (manual unguals in this research) to different functional simulations. Using FSA, the functional characters can be well-defined quantitatively and compared within a comprehensive framework against other quantified metrics such as length, proportion, shapes, time scales and ontogenies. By using this method, we first assess the function and formation of these bizarre theropod unguals under a quantitative and comprehensive framework. In our research, we simulated three different functional scenarios on unguals from 19 taxa, and we reveal surprisingly distinct functional performances of unguals from these two clades, and also provide new insights into how these highly modified manual structures evolved and developed.

## Results

**FEA, FSA, and functional divergence of unguals.** Our finite-element models (Fig. 2) show functional diversity of these unguals, not only among different dinosaurs but also showing how each ungual performs differently in the three functional simulations we test (scratch-digging function, hook-and-pull function, and piercing function, Figs. 2 and 3). By our integrated FSA method, these functional divergences can be estimated clearly and visualised (Fig. 4). For each ungual, there is a 'functional space' representing its functional performance in the three simulated functions, so making each 'functional space' a triangle. In the output FSA figures (Fig. 4 and Figs. S2–6), related geometric parameters, including coordinates of vertexes and centroids, lengths of sides and areas of these triangle hulls (Fig. 4 and Figs. S2–6), all act as specific functional indicators. The greater the dimensions of the triangle and the further apart its vertices, the greater the functional divergence between the different simulations of the claw function.

Rainbow colour gradients in these plots (from blue to red) show the general stress levels from low to high. In global view, the simulation results with lowest stress distribution fall in the quadrant II (PC1 negative and PC2 positive), those with middle-stress values between the quadrant III and IV (PC2 negative), and those with highest stress levels in the quadrant I (PC1 and PC2 both positive) (Fig. 3). In general, most simulations fall on the negative side of the PC1 axis and are symmetrically distributed along the PC2 axis. As for the centroid of each functional triangle, which represents the general stress levels among three functional scenarios, only *Therizinosaurus* is in the quadrant I; four alvarezsauroids, one therizinosaurian and *Guanlong* are in the quadrant II, together with the carnivorous mammal *Puma*; other unguals, including two alvarezsauroids, one therizinosaurian and *Allosaurus* are in the quadrant III, together with the insectivorous mammal *Manis* and extinct ground sloth *Eremotherium*; and one alvarezsauroid and three therizinosaurians fall in the quadrant IV, with the insectivorous mammal *Tamandua*.

We introduced four mammals into our simulation, three of them are extant and one is recently extinct, but all with clear independent evidence about their forelimb manual functions. Using these unguals of known function for references, we compare the simulated results from extinct dinosaur unguals and attempt to reveal similarities and differences. In our simulation of the three functions, namely piercing, pulling, and digging, we tested how unguals from these mammals performed their functions. *Puma*, the hunter, and also a good climber, has a large functional hull located in the lower stress field (Fig. 3a). *Manis*, the true digger, has an almost linear functional hull of very small area (Fig. 3a). *Eremotherium*, the ground walking herbivore[32], has a middle-sized functional hull located in the middle-stress field (Fig. 3a). *Tamandua*, the climbing anteater, has a very narrow, small-sized functional hull, ranging from the

middle (pulling and piercing) to high (digging) stress fields (Fig. 3a).

In our analyses, we also included two non-maniraptorans, *Guanlong* and *Allosaurus*, as the outgroup to Maniraptora in our comparison. As typical carnivorous theropod dinosaurs[33,34], *Guanlong* and *Allosaurus* have relatively large functional hulls. In *Guanlong*, the simulated results are all located in the lower stress field and slightly apart (Fig. 3b), very close to the topological form of *Puma* (Fig. 3a), suggesting generally good performance in all three tested functions. In *Allosaurus*, the simulation result of piercing falls into a very low-stress field close to *Guanlong*, and its pulling results are located in a slightly higher stress field than *Guanlong* (but still in the low-middle-stress field), but its unguals behaved worst in scratch-digging, reaching the worst levels in all three simulations (Fig. 3b).

We considered seven alvarezsauroids in our FSA, including three Late Jurassic representatives, two from the Early Cretaceous and two from the Late Cretaceous. Early-branching alvarezsauroids have relatively large functional hulls and lower stress level locations (Fig. 3c), some comparable to those of non-maniraptorans (Fig. 3b). Only the latest two alvarezsauroids, *Linhenykus* and *Mononykus*, have unique ungual function hulls, which are extremely small and short-linear shaped (Fig. 3c), like that of *Manis* (Fig. 3c).

Six therizinosaurians, including two from the Early Cretaceous and four from the Late Cretaceous, also were analysed by FSA. We surprisingly discovered most functional hulls of therizinosaurians are middle-sized and located in the low to middle-stress level areas, and all have digging simulation results located in the highest stress field compared to the other two functions, comparable to unguals of the ground walking herbivore *Eremotherium* (Figs. S4–5), but with only one exception, *Therizinosaurus* (Figs. S4–5). The functional hull of *Therizinosaurus* has a moderate-sized area, and unique locations in very high-stress level regions that are not occupied by any other taxa in our FSA (Figs. S4–5).

In considering the functional divergence between hook-and-pulling and piercing (Fig. 4a and Fig. S6a), most tests of unguals show negative or weakly positive results, indicating that generally the hook-and-pull function shows lower stress than piercing, or a similar stress distribution. The only exception is *Allosaurus*, for which the lowest stress is in piercing (Fig. 4a and Fig. S6a). On the contrary, most unguals test as strongly positive to weakly negative when scratch-digging and piercing are compared (Fig. 4b and Fig. S6b) indicating that piercing generated higher stresses for many taxa. The two Late Cretaceous alvarezsauroids *Linhenykus* and *Mononykus*, with their single functional fingers, show essentially no difference between these two functions, indeed high functional convergence (Fig. 4b and Fig. S6a). The result for *Tugulusaurus* is very close, but the stress is slightly higher in scratching (Fig. 4b and Fig. S6b). Most unguals show strong positive values for scratching compared to hook-and-pull, meaning that scratching generated higher stresses. Only the ungual of *Guanlong* is distinctly opposite (Fig. S6c), and the functional divergence of both one functional fingered alvarezsauroids, *Linhenykus* and *Mononykus*, are very small, being slightly negative (Fig. S6c).

When the area of each functional triangle is considered (Fig. S4), our test results go far beyond previous expectations of the functions of theropod unguals[12,15]. Unguals of the non-maniraptoran theropods *Guanlong* and *Allosaurus*, and unguals of the earliest-branching alvarezsaurid *Aorun*, have the three largest functional triangles, suggesting they have divergent and variable performance when tested in the three different functions. The unguals of the living carnivore mammal *Puma* also show a large functional hull, second only to the carnivorous theropods. The unguals from most therizinosaurians have a middle-sized functional hull area, together

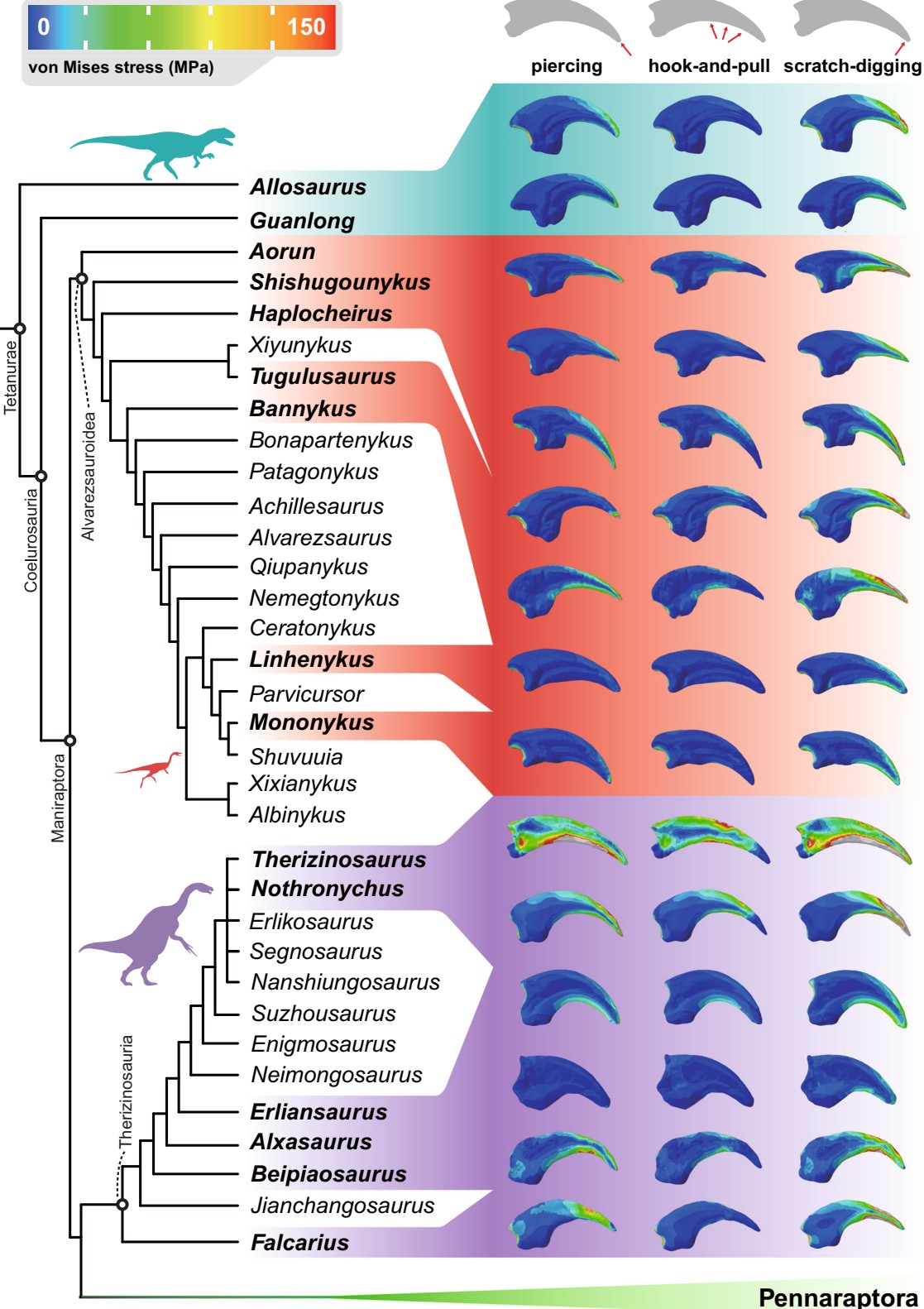

**Fig. 2 Phylogenetic comparison of von Mises stress plots of dinosaur manual unguals under three functional scenarios.** Unguals are scaled to the same surface area and illustrated in lateral view. This phylogeny is based on studies by Qin et al.[7] and Zanno et al.[66,67]

with early alvarezsauroids like *Shishugounykus* and *Tugulusaurus*. Interestingly, alvarezsauroids like *Linhenykus* and *Mononykus*, which are usually interpreted as having had specific functions of their unguals[4,14], have the smallest areas, close to zero, suggesting

that their ungual functional consistency is extremely high. These results are surprisingly opposite to the previous assumption that both alvarezsauroids and therizinosaurians had highly modified, single-function adapted unguals[4,14].

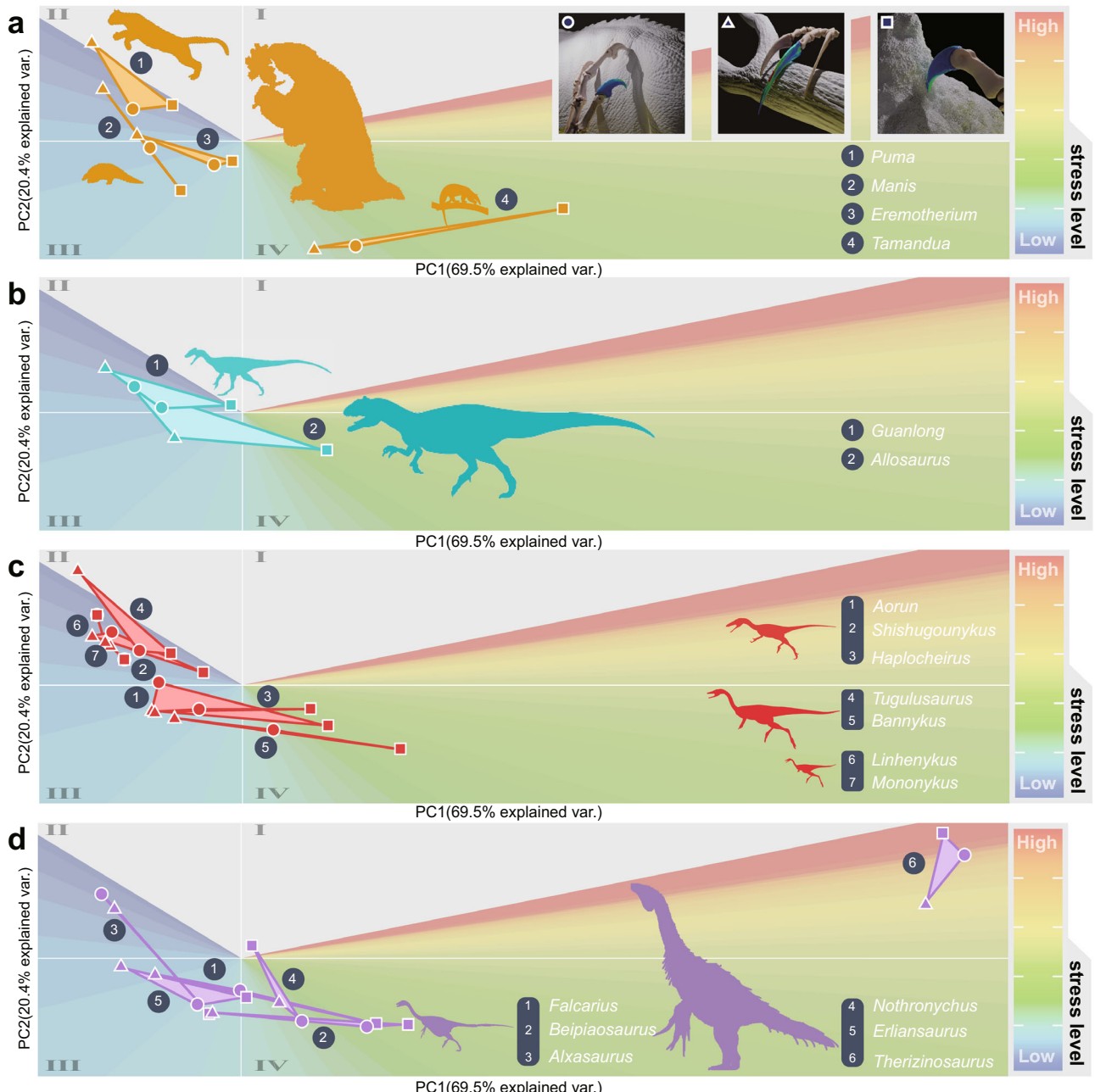

**Fig. 3 Functional space plots illustrating the divergent functional performances of manual unguals.** Unguals from reference mammals (**a**), non-maniraptoran theropods (**b**), alvarezsauroids (**c**) and therizinosaurians (**d**), tested in three major ungual functional scenarios, piercing (marked as circles, demonstrated by manus of *Guanlong*), hook-and-pull (marked as triangles, demonstrated by manus of *Therizinosaurus*) and scratch-digging (marked as squares, demonstrated by manus of *Linhenykus*). Silhouettes roughly exhibit their body forms and sizes.

In general, our FSA suggests that only late-branching alvarezsauroids and therizinosaurians had highly modified ungual functions. The late-branching alvarezsauroids *Linhenykus* and *Mononykus* show striking consistency of function either between scratch-digging and piercing or between hook-and-pull and piercing (Fig. S6c). The three Jurassic Shishugou alvarezsauroids, *Shishugounykus*, *Aorun* and *Haplocheirus*, exhibit the largest functional divergence between scratch-digging to piercing, but medium divergence between hook-and-pull to piercing (Fig. 4 and Fig. S6). By contrast, Early Cretaceous, large-sized alvarezsauroids show the largest divergence between hook-and-pull to piercing, but medium divergence between scratch-digging to piercing (Fig. 4a–c). Compared to alvarezsauroids, there is no clear trend in the evolution of therizinosaurian unguals and most

divergence lengths fall into the moderate range (Fig. 4 and Fig. S6). What is interesting is that one exception, the unguals of *Alxasaurus*, seems to have had the largest functional divergence between scratch-digging and piercing among all tested unguals, but also one of the smallest functional divergences between hook-and-pulling and piercing (Fig. 4 and Fig. S6).

**Total evidence assessment of ungual functions.** Our FSA method allows us to regress quantised functional performance to other important biological indexes, including but not limited to quantised morphological divergences (Fig. S7), ungual volume and body mass (Figs. 5 and 6). This allows us to not only quantitatively show high or low functional performances, but also

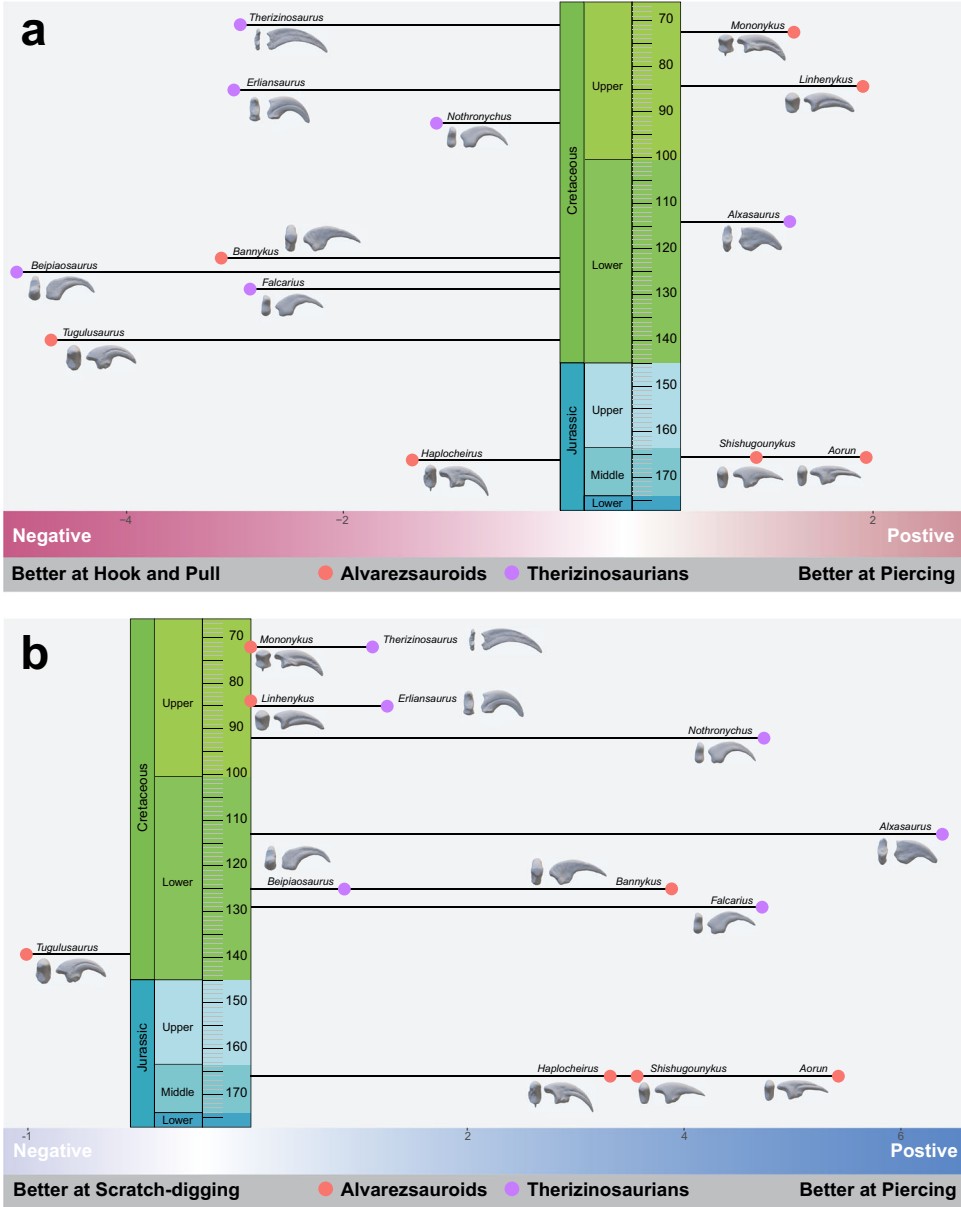

**Fig. 4 Quantified manual ungual functional divergence and overall functional performance estimation though geological time scales.** Functional divergences are exhibited between hook-and-pull and piercing (**a**), and scratch-digging and piercing (**b**) within alvarezsauroid and therizinosaurian claws.

to study the coupling relationship between structure functions, shapes, and sizes, and shed new lights on how these specialised structures are formed.

When all factors are considered together, we find that the shape specialisation of unguals generally follows the size increase of unguals and the gigantism of therizinosaurians (Fig. 5). But the evolution of functional performance of the unguals is not simply linear. Unguals from the earliest, also smallest, therizinosaurians to the middle-sized therizinosaurians show slightly increased general functional performance (the lowest average stress found in *Alxasaurus*). After the therizinosaurians started to increase in body size, the functional performance of the large-sized therizinosaurian unguals rapidly diminished.

In alvarezsauroids, the situation is more complicated. When the volume, morphological and functional divergences were regressed to body size, we can see only the ungual sizes are clearly related to body sizes (Fig. 6a). Either the functional performance

or the major morphological modification is irrelevant to body size, and even irrelevant to the size of unguals (Fig. 6a, c, d). On the contrary, the functional performances of alvarezsauroid unguals are highly related to a tendency of increasing the ratio of depth to width (Fig. 6b, Fig. S7). The length proportions of manual ungual II are higher in late-branching taxa than in the Late Jurassic Shishugou taxa, regardless of the real size of these unguals.

## Discussion

Maniraptorans were named by Jacques Gauthier in 1986, meaning 'hand snatchers', but maniraptoran hands actually had far more diverse functions[35]. The most diverse, and only extant maniraptoran clade, Aves, have reduced or modified manual unguals as parts of their wings[1], with rare exceptions in juveniles[36]. Conversely, other paravians, including dromaeosaurids and troodontids, have long and slender grasping hands, and their manual unguals are not highly specialised[37,38]. The process of ungual

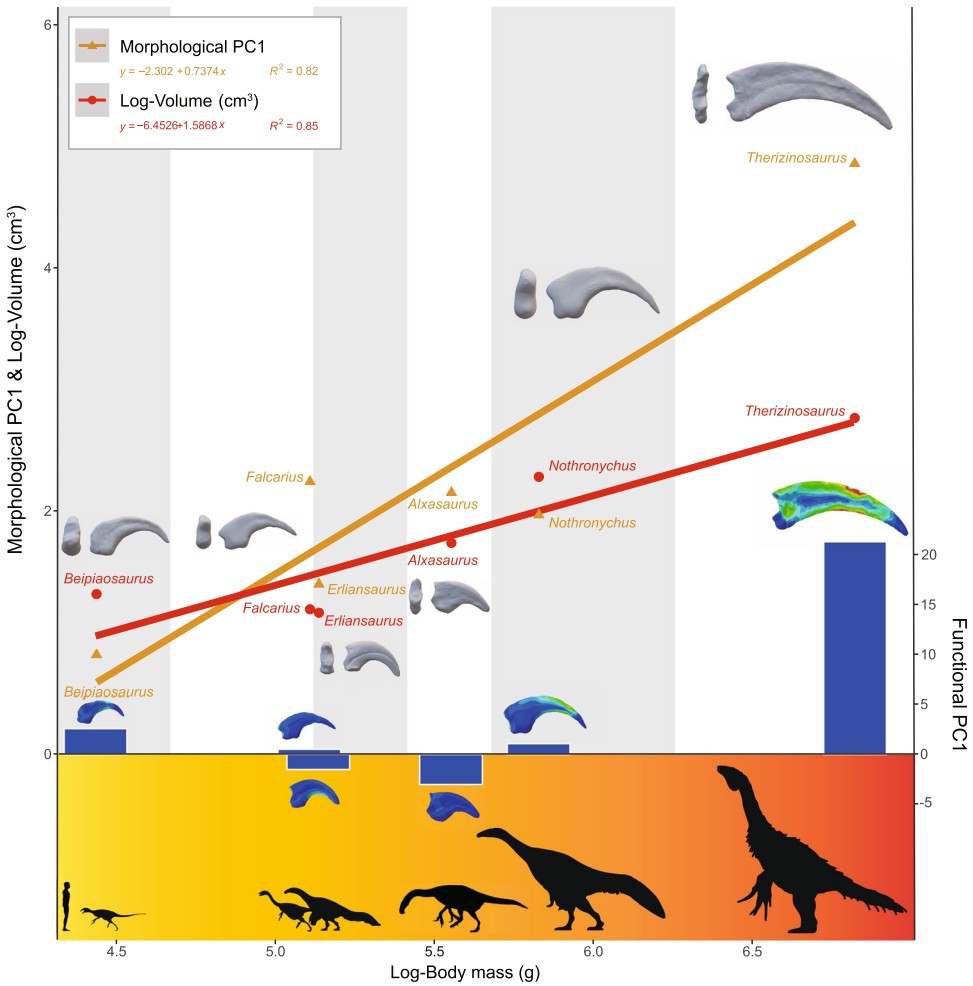

**Fig. 5 Linear regressions of morphologies, functional performances and ungual sizes vs. body size of therizinosaurians.** Histogram of functional performance showing divergent evolutionary paths of unguals from late gigantic members.

degeneration and hand modification was related to the formation of wings[1]. It is central to long-standing debates about how flight arose, whether from the ground up or the trees down[1,39], because it has influenced our understanding of whether the earliest birds like *Archaeopteryx* could climb trees or not. Manual unguals of some early-branching lineages of maniraptorans, Oviraptorosauria for example, also share the most common morphological features with those of dromaeosaurids and troodontids[40]. But most puzzling is that the two earliest-branching lineages of maniraptorans, namely the alvarezsauroids and therizinosaurians, which have highly modified and bizarre manual unguals on modified hands. Moreover, alvarezsauroids and therizinosaurians are closely related, but both lineages bear highly specialised unguals in derived members, yet ungual specialisation evolved in opposite directions through time. Late-branching therizinosaurians had elongate and sickle-like manual unguals[12], while the manual unguals of late-branching alvarezsauroids were stout and rock-pick-shaped[14,15]. Previous functional speculation about theropod unguals was mainly based on two chains of evidence, one being extensively sampled, morphologically based shape analyses, referring to living animal unguals and their functions[27,28,41,42], most of these studies were on pedal unguals, but very few other studies were clade-specific simulated FE analyses on manual unguals[30]. Both approaches have provided knowledge of functional performance by structural mechanical indicators.

Therizinosaurians are always regarded as the most bizarre theropod dinosaurs, not only because of their famous large

manual unguals, but also their large body sizes, long necks, and relatively small, sauropodomorph-like heads[19,43]. Previous hypotheses about the ecology of therizinosaurians are also controversial, including herbivory, omnivory and insectivory, and these ecological hypotheses all make corresponding assumptions about the function of their distinct forelimbs and especially their manual unguals. These assumptions include the hook-and-pull of vegetation branches in the herbivore hypothesis[44], scratch-digging of dirt and termite nests in the insectivore hypothesis[45,46], and piercing when grasping prey, attacking or defending in the omnivore hypothesis[12,44]. Our FSA showed that therizinosaurian unguals behaved worst in scratch-digging. The early-branching therizinosaurians including *Alxasaurus*, *Erliansaurus* and *Falcarius* performed well in piercing and pulling, so the assumptions of grasping and pulling at vegetation and branches, or grasping prey, are feasible.

In the specialised and giant Late Cretaceous therizinosaurian *Therizinosaurus*, simulations of all three basic manual functions show that its sickle-like unguals faced a risk of structural failure. The only previous study on biomechanics of therizinosaurian unguals showed the sickle-like unguals of *Therizinosaurus* possessed the worst functional performance[12], but it was hard to tell how bad these highly specialised structures were beyond the therizinosaurians. Our study sampled their sister lineage Alvarezsauroidea widely, as well as ancestral forms (e.g., *Allosaurus* and *Guanlong*) and mammalian unguals for comparison (Fig. 3), and the unguals of *Therizinosaurus* retain the worst performance. Our

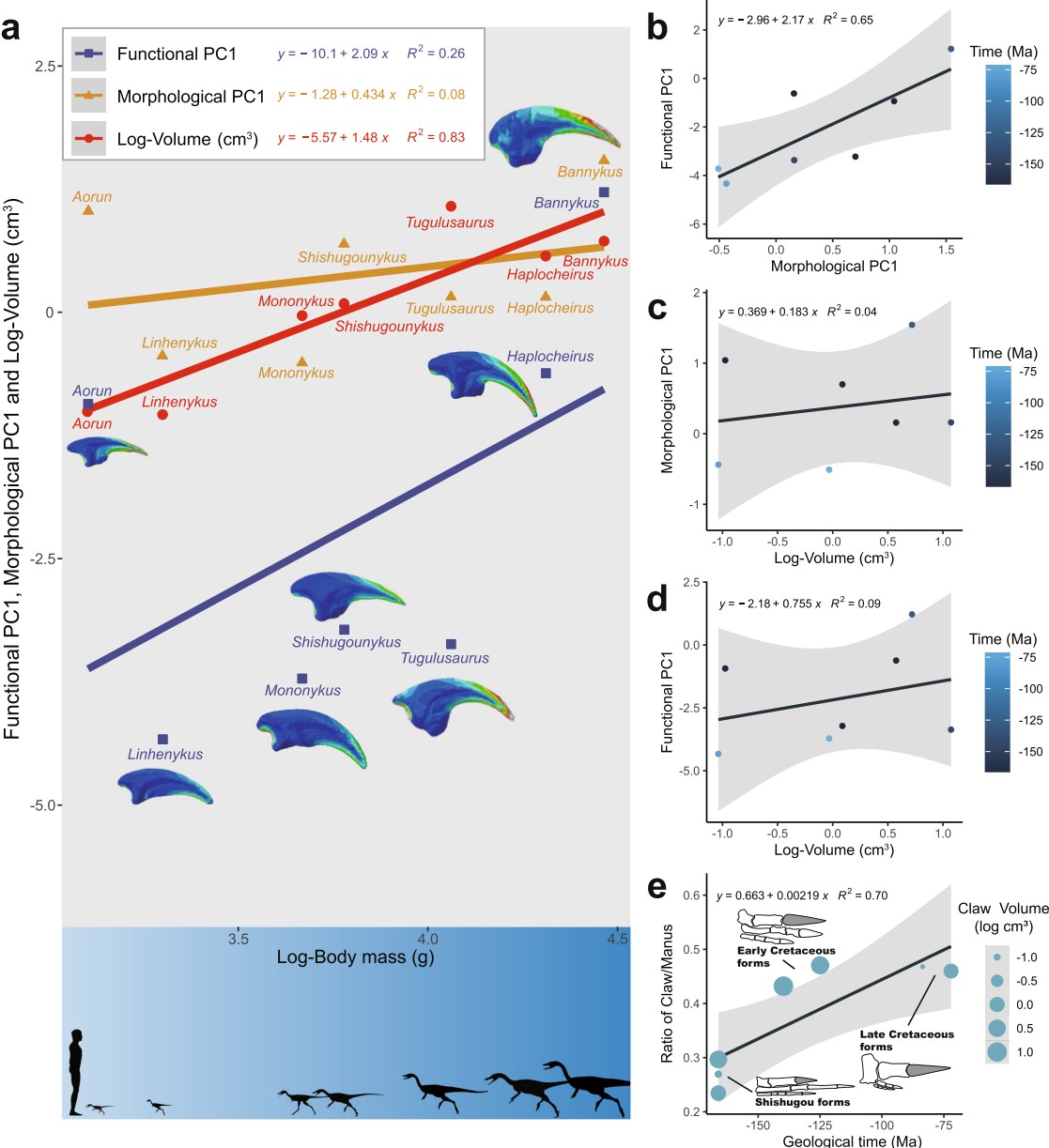

**Fig. 6 Linear regressions of average functional performance, morphologies, and ungual size vs. body size of alvarezsauroids.** (**a**) a good regression relationship only to the ungual sizes ($R^2 = 0.83$, $P$ value = 0.005 < 0.05). Linear regressions of functional divergences vs. morphological divergences (**b**), morphological divergences vs. log-transformed ungual volume (**c**), functional divergences vs log-transformed ungual volume (**d**) and proportion of length of unguals to the manus through geological time (**e**), showing close relationship between functional and morphological divergences ($R^2 = 0.65$, $P$ value = 0.029 < 0.05), but failing to support a strong relationship between functional performance and morphological specialisations to ungual volumes ($R^2 = 0.04$, $P$ value = 0.071 > 0.05; $R^2 = 0.09$, $P$ value = 0.510 > 0.05), and higher proportion of manual unguals in late-branching alvarezsauroids ($R^2 = 0.70$, $P$ value = 0.019 < 0.05).

FSA failed to support any previous functional assumptions requiring considerable stress-bearing, including digging earth[45,46], gripping and ripping branches[44], attacking prey or predators[12,44] and even climbing tress[47]. Hence, the FSA results instead suggest functions rarely requiring stress-bearing, such as exhibition, intimidation[12] or sexual display[19], or as a result of sexual selection and allometry such as in the giant antlers of *Megaloceros giganteus*[24]. According to previous anatomical investigation of forelimbs in therizinosaurians, the late-branching taxa show complex functional evolution[19], including increased dorsal reach, increased wrist flexibility, and severe reduction in manual digit length, but lengthening and mediolateral flattening manual unguals. For unguals, the descendants seem to have less

developed flexor tubercles, leading to lower mechanical advantages[48]. The differences in mechanical advantages between early to late sickle-like unguals corresponds well with our simulation of their structural strengths (Fig. 5). Our functional analyses show that therizinosaurians other than *Therizinosaurus* had overall good performance within the three functional scenarios we tested, and their manual unguals were likely involved in some motor functions of their forelimbs (Fig. 3).

The combined quantitative evidence from shape, function and measurements shows a consistent tendency of morphological specialisation along with scale increase, but according to the general functional performance we test, the *Therizinosaurus* ungual is a bizarre outlier with very high overall stress

distribution (Fig. 5), which mean very low functional performance. This could be interpreted as adopting a new specific ecological niche, which could be the long-discussed diet transformation from omnivorous to herbivorous[5,21]. Subsequent modification of the unguals from medium to giant therizinosaurians appears to have been an independent process, with the ungual shape becoming very long and narrow, finally forming the well-known sickle-like unguals (Fig. 5). At the same time, the functional performance diminishes and the unguals experience comparatively very high stress (Figs. 3 and 5). Our analyses are based on size-scaled ungual data (meaning the *Therizinosaurus* ungual is scaled to have similar size to other unguals). If we account for the large volume of its ungual and characteristics of its bony tissues, the real function could have been even poorer and very weak. We argue that the evolutionary process of forming the classic 'sickle-like unguals' was probably caused by heterochrony, namely peramorphosis[49]. The extremely enlarged, narrowed and functionally generally useless unguals of *Therizinosaurus* might have overgrown in proportion to the already very large body size, just as the huge antlers of *Megaloceros giganteus*[24], or some overly complicated frills in late ceratopsians[50] may have enlarged far beyond functional requirements.

Based on our functional simulation and analyses, the closest ecological analogue of therizinosaurians can be quantitively discussed. As previously hypothesised from functional simulation and track evidence[12,44], the early-branching therizinosaurians with non-specialised unguals could be analogues of the giant ground sloths such as *Megatherium*. Our analyses support this hypothesis and further reveal a highly analogical function to unguals of the giant ground sloth *Eremotherium*. Also, like giant ground sloths, therizinosaurians were most likely herbivores, as their body forms do not support carnivorous ecology[10], and their body size is far beyond the limits of insectivorous ecology[13]. As for the largest therizinosaurian with highly modified sickle-like manual unguals, *Therizinosaurus*, its ungual functions showed a very poor ability to bear stress. In fact, our regressions of morphological specialisation to size-related parameters show a consistent tendency, but functional performances are varied (Fig. 5). The manual unguals of *Therizinosaurus* have several orders of magnitude higher stress distributions when compared to other therizinosaurian unguals, suggesting they could hardly have functioned in as useful a manner as the other unguals, but more likely were decorative structures (Fig. 5). Our functional evidence cannot reject the ecological analogy of *Therizinosaurus* to giant ground sloths, because it could still feed efficiently by its highly modified jaws[26], and elongated neck[5]. Therefore, considering their strictly bipedal, purely terrestrial lifestyle and having the largest body size of any theropod dinosaurs, the elongated hand with sickle-like unguals might have been used mostly for threatening enemies or exhibition when mating, by analogy with the otherwise useless wings seen in ostriches.

In previous work, also extremely bizarre, single-fingered Late Cretaceous alvarezsauroids like *Linhenykus* and *Mononykus*[4,14], were said to have used their unguals for digging dirt or the nests of social insects[14,15,22,23]. In *Mononykus* for instance, its functional hull shares a very close linear topology with *Manis* (Fig. 3), the true digger and anteater. Unguals of *Mononykus* all perform equally well in piercing, scratch-digging, and hook-and-pull, with little evidence for functional specialisation. But when we look at the early-branching multi-fingered alvarezsauroids, their unguals perform poorly in scratch-digging compared to other functions. Hence, our simulation of unguals supports a potential adaptation to digging in one-fingered alvarezsauroids, showing that members of this clade built their adaptations on previous functional capabilities.

These results also lead us to rethink the innate characteristics of digging behaviour, and likely functional advantages of typical digging-adapted unguals. Digging behaviours actually require multifunctional specialisations. For example, when animals dig, they use their unguals to pierce the surface, scratch the hard surface base and pull at loose soil[51], so the functional advantage of scratching is not the sole specialisation required. At least for manual unguals, digging behaviours require unguals that can bear considerable stress whether in piercing, hook-and-pull or scratch-digging (Fig. 4). When we try to explore the possible digging habits of extinct animals, all three major ungual functions should be considered. The high functional consistency and excellent performance of unguals from late-branching alvarezsauroids and digging mammals could be interpreted as a consequence of convergent evolution, along with other previous discovered convergences, like miniaturised body size, ant-eating and nocturnality[13,52].

Based on combined quantitative evidence from shape, function, and measurements, we can also evaluate previous hypotheses on how the one-fingered manus of late-branching alvarezsauroids formed. Small body-sized alvarezsauroids generally had even smaller manual unguals (Fig. 6a), but we found no strong relationship between size variables (including the ungual size and body size) to either morphological specialisation or functional performance (Fig. 6a, c, d). But interestingly, morphological modification (in this case, increasing the ratio of the depth to width), corresponds very well to functional performance (Fig. 6b). Unguals with a high ratio of depth to width form narrower objects (Fig. S7) and behaved on average worse in our functional simulations than the wider unguals. Previous anatomical research has revealed that Early Cretaceous forms of alvarezsauroids share many of the digging-adapted features of the forelimbs but retain a plesiomorphically longer forelimb[53]. The proportion of manual ungual II contributed substantially to their hand from the Early Cretaceous onwards and continued into Late Cretaceous alvarezsauroids (Fig. 6e).

Just like other theropod clades, such as Abelisauroidea and Tyrannosauridea, which also had late-branching short forelimb forms[53,54], there are also views that forelimbs of late-branching alvarezsauroids are vestigial structures[55]. However, this hypothesis has been challenged by the fact that they share many anatomical features with digging tetrapods[14], e.g., robust humerus, ulna, and radius; fused carpometacarpus; elongation of the deltopectoral crest, olecranon process and the most distinct robust manual unguals[4,14]. It is widely understood that alvarezsauroids experienced rapid body size miniaturisation in the middle of the Cretaceous[13]. It is worth noting that the forelimbs of alvarezsauroids shorten even faster than the shrinking of body size[53], considered by some as evidence of forelimbs degradation and against the digging hypothesis[55] based on an argument that most living digging animals have long forelimbs compared to hindlimbs. In fact, the forelimbs in digging mammals having a similar length to the hindlimbs could relate to the universal quadrupedal posture in mammals. But alvarezsauroids inherited bipedal posture from early ancestral Theropoda[56], meaning that forelimb length is free from the requirement of typical quadrupedal movements. The latest hypothesis that late-branching alvarezsauroids dug damp wood termites also does not require particularly long forelimbs[13,22]. On the contrary, the shortening of the forelimbs provides unprecedented functional advantages. In lever terms, the shortening of ulna and radius shortened the out-lever arm, and the elongation of the deltopectoral crest increased attached muscle volume and promoted the input force[22]. These anatomical specialisations allow these already miniaturised late-branching alvarezsauroids to increase output force to the distal end of the forelimb tips, namely the unguals.

Our simulation shows that these highly modified unguals can bear large stresses (Figs. 3c and 4). In our research, distinct functional advantages are shared by the unguals of one-fingered alvarezsauroids (Fig. 3c), which are unlikely in a vestigial structure, supporting that these enlarged bony manual unguals (Fig. 6e) had one or more possible functions. This is a another paradigm of functional selection of morphology, comparable to the classic example of odd-toed foot evolution in horses[57].

We can identify three stages of alvarezsauroid digital evolution based on earlier work and our functional simulation results. The Jurassic Shishugou alvarezsauroids had plesiomorphic grasping hands with the longest digit III[7,53], and with manual unguals adapted to grasping and piercing, but weak in digging (Figs. 3, 4). When we further investigate details of functional advantages of the three Shishugou alvarezsauroids, our results show that the largest sized *Haplocheirus* is better at hook and pull functions, but two smaller sized species are better at piercing[7,13,58]. It should be noted that ontogenetic stages of their holotypes are different, the holotype of *Aorun zhaoi* is a hatchling, *Haplocheirus sollers* is a juvenile, and *Shishugounykus inexpectus* is a sub-adult, but the adult body size between the latter two are clearly differently by histological evidence[13,59]. Corresponding to their various early-branching body sizes, our functional indicators also support a possible ecology niche partitioning among early alvarezsauroids. Early Cretaceous alvarezsauroids such as *Bannykus* already possessed some digging adaptions, with long forelimbs but equally long manual fingers[7,53], but not all of their unguals are as structurally rigid (large functional hulls of *Tugulusaurus*, Figs. 3, 4d) as their Late Cretaceous descendants. The Late Cretaceous alvarezsauroids might have been the most specialised diggers among tetrapods, best exemplified by the monodactyl *Linhenykus*. Their forelimbs had evolved as very efficient and ideal digging tools, with large input force and short out-lever arms, and more importantly, a considerable stress-bearing manual ungual structure among tetrapods (Figs. 3, 4). Hence, we infer that late-branching alvarezsauroids showed exclusive digging function, matching a change in diet to strict insectivore and excavation of termites from damp wood[13].

Our work has established comprehensive approaches to investigate ecological niches of extinct animals by morphological modification and performance of structures adapted to physical functions. We have shown that this method is sensitive not only in clades with highly specialised unguals but could also distinguish non-specialised unguals between *Guanlong* and *Allosaurus*. It is worth mentioning that the bony unguals do not necessarily reflect the in-vivo claw morphology, and especially in the pedal talons of living birds, keratin sheathes usually covering the bony part and their morphology are highly correlated[60,61]. However, due to the limitation of fossil preservation, few of these keratin sheathes preserved, and it is not possible to determine the sheath shapes. In the previous finite-element analysis of therizinosaurians, Lautenschlager[12] tested if conservatively constructed keratin sheaths (a thin and uniform layer of keratin surrounding the bony parts) could influence functional performance of hollow and solid finite-element models. Results showed little or no decrease in stress distribution[12]. Nor can the data to date provide evidence that prominent ungual keratin sheath shapes existed in Alvarezsauroidea or in other theropod dinosaurs which might influence the analyses. Meanwhile, in consideration of the significant difference of material mechanical properties (Young's modulus 20.49 GPa of bone and 1.04 GPa of keratin sheaths)[12,30], the existence of a keratin sheath should have minor or even no influence on functional performance pattern and divergence we exhibited.

Our approach is not limited to bizarre theropod unguals but is applicable to other structures such as teeth, scales, and horns. In particular, it enables us to explore function in extinct organisms whose bizarre structures have no modern analogues and therefore cannot be interpreted simply by reference to living animals.

## Material and methods

**Data collecting**. The three-dimensional data for unguals of the 19 taxa (Fig. 2; 15 fossil unguals and 4 living animal unguals, details in supplementary Table S2) involved in this research were collected from the following sources:

(i) Published three-dimensional data of therizinosaurian unguals, from previous research articles[7,12] and personal communications from Stephan Lautenschlager (University of Birmingham, Birmingham, UK), scanned three-dimensional data of *Mononykus* forelimbs provided by Mark Norell and Congyu Yu (American Museum of Natural History, New York, US).

(ii) High-resolution computed tomography and laser scanning of published therizinosaurian, *Erliansaurus bellamanus*, fossil materials (Institute of Vertebrate Paleontology and Paleoanthropology, Beijing, China). The ungual of IVPP V4025 was scanned by a 225kv (for small skeletal elements) micro-computerised-tomography apparatus at Key Laboratory of Vertebrate Evolution and Human Origins, Chinese Academy of Science (CAS). The ungual of LH V 0002 was scanned by an Artec3D Space Spider surface scanner at the Key Laboratory of Vertebrate Evolution and Human Origins, CAS. The three-dimensional segmentation and rebuilding of these data were performed in Avizo by the author.

(iii) Online public 3D content platform Sketchfab (https://sketchfab.com/).

All newly scanned tomography slices were cropped and adjusted using the open-source image processing package Fiji[62] to similar size sizes, and then imported into the three-dimensional data visualisation and analysis software Avizo 2021.1 (Visualisation Science Group) and generated as three-dimensional ungual models.

**3D data processing**. All ungual models were first smoothed in Avizo using global smoothing tools, and then were imported into the open-source three-dimensional data visualisation and analysis software Blender (version 3.2.2, Stichting Blender Foundation, http://www.blender.org), and were smoothed regionally using the 'Sculpting' tools, and finally generated well-smoothed 3D models. Partially defective models, most are missing claw tips, are repaired in Blender with reference to the morphology of close relatives (Fig. S8)[63]. Then these 3D models were processed as 3D tetrahedral grid meshes in Avizo, avoiding generating complex and non-closed surfaces, intersections between surfaces and extreme-shaped tetrahedral elements. Once the 3D meshes met these requirements, they were ready for use in FEA analysis.

**Finite-element analysis (FEA)**. Three-dimensional FEA was performed in Abaqus (version 6.141, Dassault Systemes Simulia Corp). Manual ungual models were meshed as three-node linear triangular elements before FEA analyses. These models were scaled to the same surface area, maintaining similar numbers of finite elements between 29,323 to 42,922. The models were assigned the elastic, isotropic, and homogeneous material properties of bone, with Young's modulus of 20.49 GPa and Poisson's ratio of 0.40, following previous models of theropod unguals[12,30].

Each ungual was constrained on the articulation surface to the trochlea of the phalanx. We simulate three different functional scenarios for each ungual representing the three most common

ungual functions, according to anatomically-based theories[14,51] and previous simulations[12,30] (see Fig. 1). A total force of 200 N was applied in each scenario, half that in previous simulations[12,30], because one of our major groups of interest, the alvarezsauroids, are extremely small dinosaurs, and the average size of our ungual models is relatively smaller, requiring weaker applied forces.

(i) A scratch-digging function with the 200 N force centred on the ventral surface of the ungual tip. The scratch-digging scenario simulates animals using their unguals for digging.
(ii) A hook-and-pull function with the 200 N force spread evenly along the ventral surface of the ungual. The hook-and-pull scenario simulates animals using their unguals to pull down vegetation.
(iii) A piercing function with the 200 N force directed opposite to the ungual tip. The piercing scenario simulates animals using their unguals to attack, especially preying and fighting.

**Analysis of FEA outputs**. FEA results were summarised into a database, and von Mises stress was recorded to predict failure under ductile fracture[64]. Areas with high-stress values (red in figures) indicate the region of structural weakness, where the structure is most susceptible to failure. The intervals' method was employed to analyse our output data from FEA models in a comparative multivariate framework[31]. By the 'intervals method,' a new class of variables, referred to as "vector for stress intervals", was created to represent different intervals of stress values, in which each interval represents a sub-volume (as a percentage) of the original model and reflects a specific range of stresses[31]. The 'intervals method' can output a datagram recording the distribution of different stress ranges from low to high-stress levels, and then be used in following multivariate statistical approaches. The number of intervals required for consistency analysis varies according to different levels of structural complexity. We settled on 50 intervals here as a balance between accuracy and analysis speed, and our tests show that more than 50 intervals do not significantly improve the simulation results (Fig. S1). The output interval matrix was normalised to a correlation matrix, which better shows the relative stress distribution[31]. Subsequently, these matrixes for stress intervals were pre-processed by neglecting the highest 1% to 5% stress intervals (avoiding extreme value effects caused by anchor points in FEA and abnormal protrusions) and then log-transformed and analysed using multivariate methods such as principal component analysis (PCA) and plotted in a biplot in R[65]. The intervals method allows for more effective and quantitative comparison of FEA outputs and consequently shows a more precise distinction between their functions and biomechanical traits.

**Functional-space analysis (FSA)**. On the basis of the intervals method, we then explore the application of multifunction simulation and comparison (exemplary codes in Supplementary Data 1). In exploring independent functional scenarios, we generate functional matrixes to estimate multifunctional performance. Because our ungual models have three different loading simulations applied, each model in our test behaved differently in FE analyses and fell into different locations on the PCA plots of the interval data. Hence, for each ungual, we simulated the three possible functions (scratch digging, hook and pull, piercing) and generated a triangular-shaped function-space, termed a 'functional triangle'.

Detailed interpretation of these functional triangles shows quantitatively how the unguals behaved in our three simulations,

and the functional divergences among them (Fig. 3 and Figs. S2–6). We use the following Euclidean geometry parameters to exhibit functional information:

*The coordinates of vertexes*. The coordinates of vertexes (namely in two principal components) represent different simulation results of unguals under different testing conditions (Fig. S5).

*The area of hulls*. The area of each triangle indicates the functional divergence of the unguals. A larger area suggests a high level of functional divergence (high functional divergence means variable functional performance in different functional simulations), and a small area suggests models have relatively consistent functional performance (close in performance under different simulation conditions, Fig. S4).

*The coordinates of centroids*. Centroids are used to represent the average testing performance of each ungual in our three simulations, which assumes that the possible simulation results are distributed evenly in function space (Fig. S4).

*The length of sides*. To better show the functional divergence between results from different function simulations, the side lengths of functional triangles were quantified to describe how unguals behaved differently in functional tests under hook-and-pull, scratch-digging and piercing scenarios. We used three vectors, 'hook-and-pull to piercing', 'scratch-digging to piercing' and 'scratch-digging to hook-and-pull', to define these sides, rather than scalars because the direction of each side has its specific indication. If the terminal points of vectors are in a higher stress field than the initial points, the vectors are set as positive, indicating the former function is less suited to claw shape, otherwise vectors are negative, indicating that the former function is better adapted (Fig. 4a–c). In this case, we assume that higher stress means claws are less suited for a particular function, as comparatively weaker loads would lead to structural failure.

These parameters mentioned above allow us to estimate and compare the functions of unguals in quantitative ways, visually showing their pros and cons, and indicating which are the multifunctional unguals and which have only some specific functions.

**Total evidence assessment of ungual function and its evolution**. The FSA method allows us to quantify the functional characteristics of different ungual models, just as we quantify and compare morphological characteristics, and further study the relationship between morphological modification and functional performance. Because our functional simulation work is based on 3D models, we also use indices of three-dimensional shape information of claws, including the ratios length/width of the claw (L/W), length/depth (L/D) and depth/width (D/W) (Table S3). These morphological data of unguals were put into a published, widely sampled manual ungual dataset of 186 living animals[42], to exhibit their 3D morphological divergence (exemplary codes in Supplementary Data 2). It should be noted here that the only application of this morphospace is to quantify the morphological character divergence and does not yield any functional or ecological implications.

Our 3D ungual models provide claw volume information (Table S3), whereas estimates of body sizes of these two lineages are already well established[10,13]. Together with other measurement data[7,53], we can comprehensively assess the evolution of these most bizarre theropod manual unguals. We regressed the quantified functional performance via morphological variation, ungual size and body size (Fig. 5 and Fig. 6), showing how

function changed though shape modification and size enlargement, and to shed new light on their possible ontogenetic growth and modification.

**Reporting summary**. Further information on research design is available in the Nature Portfolio Reporting Summary linked to this article.

## Data availability

All 3D models and CAE files involved in this research are stored at the data.bris Research Data Repository, and can be visited via https://doi.org/10.5523/bris.27ui6v0q9a69y2ekvtrdueserp. Other data source tables and supplementary figures can be found at supplementary tables.

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

## Acknowledgements

We thank Xing Xu and Qi Zhao (Institute of Vertebrate Paleontology and Paleoanthropology), and Congyu Yu (American Museum of Natural History) for their help with data collecting in China and the United States. We thank Shuyang Zhou for the 3D modelling and functional scenario restoration. We thank Tom Stubbs and Rui Ying for discussions on the methodology of statistical analyses and for improving our codes. We thank Liz Martin-Silverstone for her help with CT scanning and 3D modelling. We thank Antonio Ballell Mayoral and Melisa Morales Garcia for discussions on FE methods and research results. We thank Stephan Lautenschlager (University of Birmingham) for providing essential models and data. We thank Jordi Marcé Nogué (Universitat Rovira i Virgili) for discussions on the intervals' method. We thank Albert Chen (University of Cambridge) and Jorge Gustavo Meso (National University of Río Negro) for discussions on alvarezsauroids ecology. We thank Liz Freedman Fowler and Stephan Lautenschlager for their careful work and thoughtful suggestions that have helped improve our paper substantially. This study was supported by the Middlemiss Fund of the Geologists' Association and the Jurassic Foundation to Z.Q., European Research Council Advanced Grant 'Innovation' (ERC 788203) and NERC grant NE/I027630/1 to M.J.B.

## Author contributions

Z.Q., M.J.B or E.J.R. designed and conceived the research. Z.Q. and C.L. collected the data from fossils and extant animals. Z.Q. wrote the R code and performed the morphological and functional analyses. All authors contributed to the interpretation and discussion of results and draughted the manuscript.

## Competing interests

The authors declare no competing interests.
