## [Peer Review File · Communications Biology]

Reviewers' comments:

Reviewer #1 (Remarks to the Author):

The manuscript provides a thorough analysis of the functional morphology of the manual unguals of two unusual theropod groups (therizinosaurians and alvarezsauroids) using a combination of finite element analysis and morphometric approaches. This is an interesting topic as the unusual morphology of the unguals/claws in these two groups has been the basis for substantial discussion and speculation in the past. Although similar analyses have been conducted previously, this study provides a larger sample size and somewhat different focus making it new and a worthwhile contribution. Overall, the study has been conducted thoroughly, using appropriate techniques and metrics/statistics to quantify the biomechanical behaviour. The manuscript is well-written, has for most parts sufficient details, but lacks some important information (e.g. particularly in the methods and in several figures). The major claims are largely supported by the results, but I would generally suggest a more cautious approach to some of the statements made in the discussion. Apart from this, there are a few points where clarification or more detail are required (see specific points below).

In summary, this is an interesting contribution requiring only some more moderate modifications before it can be considered for publication.

Specific points:

Page 2/49: Zanno and Makovicky (2013, cited also in the text) found no evidence for a trend towards gigantism so I would rephrase this sentence here to correctly reflect that some therizinosaurians evolved a large body size but this is not true for the entire group.

Page 2/56: Similar to the above, the massively enlarged sickle-like unguals are not present in all members of the group. Best to rephrase to clarify this.

Page 5/109: Here and in the rest of the result, use the Roman numerals when referring to the different quadrants as in the corresponding figure for easier identification.

Page 7/194: This is interesting. This is one of the few examples where there are a few species from the same formation. Could the results in this context indicate some form of niche partitioning to avoid competition? I would like to see some discussion of this later on in the text.

Page 10/260: Please distinguish between studies using manual and pedal unguals. Most of the ones referred to here are the latter.

Page 10/267: I would disagree that the heads of therizinosaurians are sauropod-like. Maybe sauropodomorph-like but maybe just remove the comparison completely.

Page 10/268: use "herbivory, omnivory, and insectivory"

Page 10/276: Given that Lautenschlager (2014) is the only study on therizinosaur claw fea/biomechanics some more comparisons between the results would be useful. The sample size in your study is a bit bigger and has a different quantitative approach which could be compared.

Page 11/289: It would be useful here to discuss that the bony unguals do not necessarily reflect the in-vivo claw morphology. A keratin sheath covering the bone may or may not reflect the underlying morphology. I appreciate that there is limited information on the final shape but this would impact on some of the inferences made here.

Page 11/311ff: This section is a bit too interpretive for my taste. Firstly, the regression cited as evidence is problematic as it is only done on a subsection of the data (see specific comment on the corresponding figure). Secondly, the unguals of *Therizinosaurus* perform worse in a comparative context. Given the inherent uncertainties of FEA we cannot get absolute values (including material yield, etc.) for the models. Therefore, I would make the statements about decorative/display structures less definitive.

Page 13/388: "most stress-bearing structure ever" is quite a statement. This would require a

similar analysis of all unguals in tetrapods, so I would tone down the hyperbole here.

Page 13/Data collecting: This section requires a lot more information about (1) specimens used (including repositories and museum numbers), (2) digitisation method, (3) resolution and associated information for CT and surface data. Further, are these models available as part of the publication? There is no data availability statement and reference to data requests are not sufficient.

Page 13/428: Where do these values for the material properties come from?

Page 14/429ff: How were these models constrained?

Figure 1: This is a good overview on the methodological process but labelling the individual steps would be useful to identify the techniques and methods clearly.

Figure 3: Overall, I would suggest making parts A-D page width to bring out the details more clearly. As it is, the individual parts are difficult to recognise. The illustrative examples below can be a bit smaller if necessary. In A, it would be useful to label the individual taxa with their corresponding functions used as a comparative reference. In the caption, explain the Roman numerals to make the identification of the quadrants discussed in the text easier.

Figure 4: Label the x-axis. It is unclear what is shown here.

Figure 5: This figure is problematic. Why are two regressions selected for therizinosaurians? Therizinosaurus is clearly an outlier reflecting its unusual morphology but either disregard for the regression or include it and do a single regression for all taxa as in figure 6. This seems to be too much like cherry-picking.

Reviewer #2 (Remarks to the Author):

This paper uses new variations on Finite Element Analysis to examine physical stresses on the bizarre claws of alvarezsaur and therizinosaur dinosaurs. The results do show some specialization toward certain behaviors, but the complex analyses run the risk of overinterpreting from limited data. After various minor edits, this should be suitable for publication.

This paper will be of interest broadly to vertebrate paleontologists, even those that do not work on these particular theropods. The paper may or may not have lasting influence on the field of theropod biomechanics.

You need a specimen list that clearly explains if and how any claws were reconstructed to have more complete tips. The claws in your analysis with complete pointy tips performed worse at scratch digging, whereas blunter claws were more stable during scratch digging. However, are the derived claws (*Linhenykus* and *Mononykus*) truly less pointy, or are they simply not reconstructed that way?

The recently described Hell Creek alvarezsaur *Trierarchunchus* demonstrates that latest Cretaceous alvarezsaurids had sharp pointed tips to their claws, with prominent curvature. If you included this claw in your analysis, what results would you expect to find? The *Mononykus* holotype claw is missing its tip; perhaps the tip should have been reconstructed as pointed as that of *Trierarchunchus*.

Fowler et al 2020. *Trierarchunchus prairiensis* gen. et sp. nov., the last alvarezsaurid: Hell Creek Formation (uppermost Maastrichtian), Montana. *Cretaceous Research* 116:104560
<https://www.sciencedirect.com/science/article/abs/pii/S0195667120302469>

Most of the alvarezsaur claws did not perform equally well at all three motions; they were generally best at hook-and-pull and worst at scratch-digging. This difference is most notable in the claws with the sharpest tips, another reason why tip reconstruction is important to the results.

Additional comments:

Line 45 typo- "studied in detailed" should be "studied in detail".

Line 49: "Pickaxe" is a strange term to describe alvarezsaur unguals – a pickaxe is pointed but long and skinny with little curvature. An alvarezsaur ungual is short, stout, and notably more curved than a pickaxe.

Define use of "hook and pull" – some authors use this to mean piercing/impaling followed by pulling or levering, while others use it to mean grabbing by looping the claw tip around and behind an object, then pulling.

Line 100: What are the "three simulated functions"? Assume the reader hasn't scrolled down to the methods or Figure 2 yet and list them briefly here.

Figure 4: The font size of the taxon names is extremely small and illegible unless it is magnified several times. Will the published version be large enough and hi-res enough that most readers could read the names?

Line 179-180: "Our test results go far beyond previous expectations" – what were the previous expectations? Citation?

Line 201 "Alxasaurus....between snatching and piercing" – you haven't used the term snatching before; it isn't one of the three functions you've been listing.

Figure S6 was the easiest to understand your results and conclusions, compared to all other figures. This may be more useful in the main manuscript. However, it also highlights that the results within therizinosaurs and within alvarezsaurians are rather all over the place, suggesting both clades have similar skills at hook-and-pull and piercing. Why would such wildly different claw morphologies give similar results? It is interesting, but not surprising, that both groups are poor at scratch-digging. Scratch-digging claws are generally broader (e.g. tortoise).

Figure S7: Some of the silhouettes are unidentifiable. What are the two blue subterranean blobs? Maybe one is a mole, but what is the upper one? What is the yellow arboreal blob on the right? The cat and sloth are much more obvious.

Is figure S7 implying that therizinosaurs are arboreal like ground sloths and some cats?

Line 207: "allow us" should be "allows us".

Line 305-306: "closest ecological analogue of therizinosaurs is clear" Might be overstating a bit. Yes, therizinosaurs overlap with ground sloths in functional space as shown in Figure 3, but so do alvarezsaurians, with their vastly different claw shape and size.

Methods line 428: You used the material properties of bone, but claws would have been covered with a keratin sheath that makes the claw overall larger and more curved, as well as absorbing some of the stress before it reaches the bone. How do the material properties of keratin affect the analysis? You need to discuss how it works in modern animals and how your bone-only model might change if the size and material properties of keratin were included.

Lines 476-477: "A larger area suggests a high level of functional divergence, and a small area suggests relatively consistent functions." – What do you mean by "consistent functions"? Do you mean that a claw had similar FEA performance in all three simulations? So it did all three functions at a similar level of success? Does "functional divergence" mean that it is good at diverse functions, or that it performed well in some functions and terribly in others?

After these various minor edits, this should be suitable for publication.

-Dr. Liz Freedman Fowler

Responses to the referees

Responses to Reviewer #1 (Remarks to the Author):

The manuscript provides a thorough analysis of the functional morphology of the manual unguals of two unusual theropod groups (therizinosaurians and alvarezsauroids) using a combination of finite element analysis and morphometric approaches. This is an interesting topic as the unusual morphology of the unguals/claws in these two groups has been the basis for substantial discussion and speculation in the past. Although similar analyses have been conducted previously, this study provides a larger sample size and somewhat different focus making it new and a worthwhile contribution. Overall, the study has been conducted thoroughly, using appropriate techniques and metrics/statistics to quantify the biomechanical behaviour. The manuscript is well-written, has for most parts sufficient details, but lacks some important information (e.g. particularly in the methods and in several figures). The major claims are largely supported by the results, but I would generally suggest a more cautious approach to some of the statements made in the discussion. Apart from this, there are a few points where clarification or more detail are required (see specific points below). In summary, this is an interesting contribution requiring only some more moderate modifications before it can be considered for publication.

Many thanks for these positive comments. In the revised manuscript, we added more technical details and data source information. Based on your suggestion, we have adjusted the expression of some viewpoints, in order to be more cautious. Please see details in the Specific points.

Specific points:

Page 2/49: Zanno and Makovicky (2013, cited also in the text) found no evidence for a trend towards gigantism so I would rephrase this sentence here to correctly reflect that some therizinosaurians evolved a large body size but this is not true for the entire group.

This advice was followed. We re-write this part as 'some members of the therizinosaurians evolved large body sizes'.

Page 2/56: Similar to the above, the massively enlarged sickle-like unguals are not present in all members of the group. Best to rephrase to clarify this.

This advice was followed. We re-write this part as 'With reference to the most remarkable elongate sickle-like unguals emerged in late-branching members'.

Page 5/109: Here and in the rest of the result, use the Roman numerals when referring to the different quadrants as in the corresponding figure for easier identification.

This advice was followed.

Page 7/194: This is interesting. This is one of the few examples where there are a few species from the same formation. Could the results in this context indicate some form of niche partitioning to avoid competition? I would like to see some discussion of this later on in the text.

Many thanks for these positive comments on this discovery, and this advice we followed. We add more discussion about these three early-branching alvarezsauroids, referring their body size

variation, functional divergence and possible niche partitioning.

Page 10/260: Please distinguish between studies using manual and pedal unguals. Most of the ones referred to here are the latter.

This advice was followed. We distinguished studies using manual and pedal unguals, but we still keep all these references, because we think even the pedal unguals are instructive to our understanding of the ungual functions.

Page 10/267: I would disagree that the heads of therizinosaurians are sauropod-like. Maybe sauropodomorph-like but maybe just remove the comparison completely.

Thanks for reminding, this advice was followed. We use 'sauropodomorph-like' to describe their heads.

Page 10/268: use "herbivory, omnivory, and insectivory"

Thanks for this correction, this advice was followed.

Page 10/276: Given that Lautenschlager (2014) is the only study on therizinosaur claw function/biomechanics some more comparisons between the results would be useful. The sample size in your study is a bit bigger and has a different quantitative approach which could be compared.

This advice was followed. We add more discussion about our new understanding benefitting from our larger sampling size and new quantitative approaches.

Page 11/289: It would be useful here to discuss that the bony unguals do not necessarily reflect the in-vivo claw morphology. A keratin sheath covering the bone may or may not reflect the underlying morphology. I appreciate that there is limited information on the final shape but this would impact on some of the inferences made here.

This advice was followed. We add more discussion about the keratin sheath and its possible influence on our understanding of claw morphology functions in the final paragraph.

Page 11/311ff: This section is a bit too interpretive for my taste. Firstly, the regression cited as evidence is problematic as it is only done on a subsection of the data (see specific comment on the corresponding figure). Secondly, the unguals of *Therizinosaurus* perform worse in a comparative context. Given the inherent uncertainties of FEA we cannot get absolute values (including material yield, etc.) for the models. Therefore, I would make the statements about decorative/display structures less definitive.

This advice was followed. We agree that two-stage linear regressions are not the best choices to show the functional performance of these therizinosaurian unguals. As an alternative, we use a histogram to show the functional performance divergence, please see the new figure.5. Based on the new figure, we reorganized this section of discussion, mainly show the functional divergence via consistent morphology to size data. We follow this advice and make the statements about decorative/display structures less definitive.

Page 13/388: "most stress-bearing structure ever" is quite a statement. This would require a similar analysis of all unguals in tetrapods, so I would tone down the hyperbole here.

This advice was followed. As the advice above, we toned down our discussion here.

Page 13/Data collecting: This section requires a lot more information about (1) specimens used (including repositories and museum numbers), (2) digitisation method, (3) resolution and associated information for CT and surface data. Further, are these models available as part of the publication? There is no data availability statement and reference to data requests are not sufficient.

This advice was followed, we add more essential information on Method and tables in Supplementary materials.

Page 13/428: Where do these values for the material properties come from?

We add references and data resources here.

Page 14/429ff: How were these models constrained?

We add detail on the constraint condition of models here.

Figure 1: This is a good overview on the methodological process but labelling the individual steps would be useful to identify the techniques and methods clearly.

This advice was followed. We add more labels to make the workflow clearer.

Figure 3: Overall, I would suggest making parts A-D page width to bring out the details more clearly. As it is, the individual parts are difficult to recognise. The illustrative examples below can be a bit smaller if necessary. In A, it would be useful to label the individual taxa with their corresponding functions used as a comparative reference. In the caption, explain the Roman numerals to make the identification of the quadrants discussed in the text easier.

This advice was followed. We made a new Figure 3, and made parts A-D page width, with smaller illustrative examples.

Figure 4: Label the x-axis. It is unclear what is shown here.

After considering comments from both reviewers, we use the old Figure S6 to replace the old Figure 4. And in the new Figure 4, we exhibited the functional divergence of both clades along geological time bar. We hope this new figure can provide more information and patterns.

Figure 5: This figure is problematic. Why are two regressions selected for therizinosaurians? Therizinosaurus is clearly an outlier reflecting its unusual morphology but either disregard for the regression or include it and do a single regression for all taxa as in figure 6. This seems to be too much like cherry-picking.

This advice was followed. Please see above, we have modified the Figure 5 and also adjust the related discussion.

Responses to Reviewer #2 (Remarks to the Author):

This paper uses new variations on Finite Element Analysis to examine physical stresses on the bizarre claws of alvarezsaur and therizinosaur dinosaurs. The results do show some specialization toward certain behaviors, but the complex analyses run the risk of overinterpreting from limited data. After various minor edits, this should be suitable for publication.

This paper will be of interest broadly to vertebrate paleontologists, even those that do not work on these particular theropods. The paper may or may not have lasting influence on the field of theropod biomechanics.

You need a specimen list that clearly explains if and how any claws were reconstructed to have more complete tips. The claws in your analysis with complete pointy tips performed worse at scratch digging, whereas blunter claws were more stable during scratch digging. However, are the derived claws (*Linhenykus* and *Mononykus*) truly less pointy, or are they simply not reconstructed that way?

The recently described Hell Creek alvarezsaur *Trierarchuncus* demonstrates that latest Cretaceous alvarezsaurs had sharp pointed tips to their claws, with prominent curvature. If you included this claw in your analysis, what results would you expect to find? The *Mononykus* holotype claw is missing its tip; perhaps the tip should have been reconstructed as pointed as that of *Trierarchuncus*.

Fowler et al 2020. *Trierarchuncus prairiensis* gen. et sp. nov., the last alvarezsaurid: Hell Creek Formation (uppermost Maastrichtian), Montana. Cretaceous Research 116:104560 <https://www.sciencedirect.com/science/article/abs/pii/S0195667120302469>

Most of the alvarezsaur claws did not perform equally well at all three motions; they were generally best at hook-and-pull and worst at scratch-digging. This difference is most notable in the claws with the sharpest tips, another reason why tip reconstruction is important to the results.

Many thanks for these positive comments. We also hope that our approach can provide a new tools to understanding function evolution. For the concern of point tips of alvarezsauroid claws, we realized that it seems that our model restoration works are not detailed in the Methods section. The *Mononykus* and *Linhenykus* holotypes both missed their tip somehow, thus we already reconstructed its pointed tip, referring the unguals of *Trierarchuncus* (Supplementary Table Fig. 8 show the comparison of our reconstructed unguals of *Mononykus* to *Trierarchuncus*). In the new manuscript, we proved more details on how to make complete unguals models and added tips by reference of the paper of *Trierarchuncus* (Supplementary Table S2 and Fig. 8). Besides, we need to explain that our method is not sensitive to abnormal protrusion on testing models. The FSA and intervals' method are designed to ignore the highest-stress distribution within a certain proportion (usually top 1% to 5% of the highest-stress distribution area). The reason why we neglect the highest-stress area is that these highest-stress area where are normally anchor points in FEA simulations, or abnormal protrusion (For example, small bumps or spikes). This approach allows us to focus more on the global stress distribution rather than exceptional values. Hence, the concern of the pointed tips does not significantly affect our functional analysis results. Please see details in the Specific points.

Additional comments:

Line 45 typo- “studied in detailed” should be “studied in detail”.

Thanks, this advice was followed.

Line 49: “Pickaxe” is a strange term to describe alvarezsaur unguals – a pickaxe is pointed but long and skinny with little curvature. An alvarezsaur ungual is short, stout, and notably more curved than a pickaxe.

This advice was followed. We use ‘rock-pick-like’ to replace the ‘Pickaxe’. We hope this tool is a better comparison for claws of alvarezsauroids.

Define use of “hook and pull” – some authors use this to mean piercing/impaling followed by pulling or levering, while others use it to mean grabbing by looping the claw tip around and behind an object, then pulling.

This advice was followed. We added our definition of ‘hook and pull’ function.

Line 100: What are the “three simulated functions”? Assume the reader hasn’t scrolled down to the methods or Figure 2 yet and list them briefly here.

This advice was followed. We list these three functions here.

Figure 4: The font size of the taxon names is extremely small and illegible unless it is magnified several times. Will the published version be large enough and hi-res enough that most readers could read the names?

Thanks for this advice. After considering comments from both reviewers, we use the old Figure S6 to replace the old Figure 4. And in the new Figure 4, we exhibited the functional divergence of both clades along geological time bar. We hope this new figure can provide more information and patterns.

Line 179-180: “Our test results go far beyond previous expectations” – what were the previous expectations? Citation?

This advice was followed. We add more references about the previous hypotheses here.

Line 201 “Alxasaurus....between snatching and piercing” – you haven’t used the term snatching before; it isn’t one of the three functions you’ve been listing.

Thanks for this advice. We wanted to refer snatch-digging here. It has been modified in the new manuscript.

Figure S6 was the easiest to understand your results and conclusions, compared to all other figures. This may be more useful in the main manuscript. However, it also highlights that the results within therizinosaurs and within alvarezsaurids are rather all over the place, suggesting both clades have similar skills at hook-and-pull and piercing. Why would such wildly different claw morphologies give similar results? It is interesting, but not surprising, that both groups are poor at scratch-digging. Scratch-digging claws are generally broader (e.g. tortoise).

Thanks for this advice. After considering comments from both reviewers, we use the old Figure S6 to replace the old Figure 4.

Figure S7: Some of the silhouettes are unidentifiable. What are the two blue subterranean blobs? Maybe one is a mole, but what is the upper one? What is the yellow arboreal blob on the right? The cat and sloth are much more obvious.

Thanks for this advice. We use clear silhouettes from Jenkins et al. 2020 (Figure 6) to replace our unclear silhouettes.

Is figure S7 implying that therizinosaurs are arboreal like ground sloths and some cats?

Thanks for this questions. We did not imply any ecological analogues by this figure, but only use this figure to show morphological divergence.

In the methods part, we explained that ‘It should be noted here that the only application of this morphospace is to quantify the morphological character divergence and does not yield any functional or ecological implications’. In this research we find some morphologies of claws from theropod dinosaurs are not falling in any morphospace composited by living references. There are limitations to using morphological data to speculate on the function of the claw. This is why we are taking our functional studies further. But morphological data are important, especially quantified morphological data that can be used in our total evidence assessment of unguis functions in conjunction with studies of functional research outputs.

Line 207: “allow us” should be “allows us”.

Thanks for this advice. This advice was followed.

Line 305-306: “closest ecological analogue of therizinosaurs is clear” Might be overstating a bit. Yes, therizinosaurs overlap with ground sloths in functional space as shown in Figure 3, but so do alvarezsaurids, with their vastly different claw shape and size.

Thanks for this advice, We toned down our discussion here. We noticed that our results can only improve understanding of possible ecological analogues of therizinosaurs and support some previously raised hypothesis and reject others.

Methods line 428: You used the material properties of bone, but claws would have been covered with a keratin sheath that makes the claw overall larger and more curved, as well as absorbing some of the stress before it reaches the bone. How do the material properties of keratin affect the analysis? You need to discuss how it works in modern animals and how your bone-only model might change if the size and material properties of keratin were included.

Thanks for this advice. In considering that both reviewers concern the keratin sheath of claws, we add a paragraph of throughout discussion about this in the final part of this manuscript. We discussed the previous simulated keratin sheath models, and other research who argued the more prominent keratin sheaths in birds pedal claws. We also explain that given the limitations of our data set, we can only use relatively conservative estimates and consider the impact of this factor to be insignificant.

Lines 476-477: “A larger area suggests a high level of functional divergence, and a small area suggests relatively consistent functions.” – What do you mean by “consistent functions”? Do you mean that a claw had similar FEA performance in all three simulations? So it did all three functions at a similar level of success? Does “functional divergence” mean that it is good at diverse functions, or that it performed well in some functions and terribly in others?

Thanks for this questions. We added more details and explanations about the ‘terms’ we used in Methods part, please see the new manuscript. We hope we've explained it well enough.

REVIEWERS' COMMENTS:

Reviewer #1 (Remarks to the Author):

I would like to thank the authors for addressing my previous comments and suggestions so thoroughly. All my questions and comments have been answered satisfactorily. The manuscript looks very good now and I have no further comments.

Reviewer #2 (Remarks to the Author):

The authors did an excellent job of incorporating the reviewers' comments. I include some very minor edits below. Otherwise, the manuscript is now suitable for publication.

page 5/line 205: "Snatch-digging" I think you mean scratch-digging.

7/277: This new sentence on the Lautenschlager reference does not actually cite the reference.

9/394-397: This new sentence on ontogeny of Aorun, Haplocheirus, and Shishigounykus is interesting, but the English phrasing is difficult to understand.

-Dr. Liz Freedman Fowler

Responses to the referees

Responses to Reviewer #1 (Remarks to the Author):

I would like to thank the authors for addressing my previous comments and suggestions so thoroughly. All my questions and comments have been answered satisfactorily. The manuscript looks very good now and I have no further comments.

On behalf of the authors, I would like to express our sincere gratitude to the reviewers. Many thanks to the reviewers for their positive comments and careful review of the manuscript.

Responses to Reviewer #2 (Remarks to the Author):

The authors did an excellent job of incorporating the reviewers' comments. I include some very minor edits below. Otherwise, the manuscript is now suitable for publication.

Thank you very much for your positive comments and careful revision. Your suggestions have contributed greatly to improving this manuscript.

Additional comments:

page 5/line 205: "Snatch-digging" I think you mean scratch-digging.

This advice was followed. We replaced it as 'scratch-digging'.

7/277: This new sentence on the Lautenschlager reference does not actually cite the reference.

This advice was followed. We added this reference there.

9/394-397: This new sentence on ontogeny of Aorun, Haplocheirus, and Shishigounykus is interesting, but the English phrasing is difficult to understand.

This advice was followed. We have modified the expression of this sentence to make it more understandable.